# Highlights on the Role of Galectin-3 in Colorectal Cancer and the Preventive/Therapeutic Potential of Food-Derived Inhibitors

**DOI:** 10.3390/cancers15010052

**Published:** 2022-12-22

**Authors:** Anna Aureli, Manuela Del Cornò, Beatrice Marziani, Sandra Gessani, Lucia Conti

**Affiliations:** 1CNR Institute of Translational Pharmacology, Via Carducci 32, 67100 L’Aquila, Italy; 2Center for Gender-Specific Medicine, Istituto Superiore di Sanità, Viale Regina Elena 299, 00161 Rome, Italy; 3Emergency and Urgent Department, Sant’Anna University Hospital, Via A. Moro, 8, 44124 Cona, Italy

**Keywords:** galectin-3, colorectal cancer, tumor microenvironment, immunosuppression, intestinal inflammation, bioactive food compounds, non-digestible carbohydrates, polyphenols

## Abstract

**Simple Summary:**

Colorectal cancer (CRC) incidence is increasing worldwide and represents an important health problem.Therapy failure and progression to metastatic disease are major concerns. Among the factors involved in tumor growth, galectin-3 (Gal-3) plays an important role due to its ability to finetune a number of molecular players that act at different levels of cancer-related processes. A clear relationship between Gal-3 and CRC has been demonstrated. Several studies have, indeed, reported a pathogenetic role for this protein in intestinal inflammation and CRC onset/progression. Moreover, some plant-source food-derived bioactive compounds (mostly fibers and polyphenols) can contribute to the control of CRC onset/growth through their capacity to block Gal-3 activities. In this review, we summarize these studies, highlighting the influence of Gal-3 on CRC risk/progression, cancer cell spreading and patient prognosis, as well as the potential of natural food-derived Gal-3 inhibitors as promising candidates for CRC prevention and therapy.

**Abstract:**

Colorectal cancer (CRC) is a leading cause of death worldwide. Despite advances in surgical and therapeutic management, tumor metastases and resistance to therapy still represent major hurdles. CRC risk is highly modifiable by lifestyle factors, including diet, which strongly influences both cancer incidence and related mortality. Galectin-3 (Gal-3) is a multifaceted protein involved in multiple pathophysiological pathways underlying chronic inflammation and cancer. Its versatility is given by the ability to participate in a wide range of tumor-promoting processes, including cell–cell/cell–matrix interactions, cell growth regulation and apoptosis, and the immunosuppressive tumor microenvironment. This review provides an updated summary of preclinical and observational human studies investigating the pathogenetic role of Gal-3 in intestinal inflammation and CRC, as well as the potential of Gal-3 activity inhibition by plant-source food-derived bioactive compounds to control CRC onset/growth. These studies highlight both direct and immuno-mediated effects of Gal-3 on tumor growth and invasiveness and its potential role as a CRC prognostic biomarker. Substantial evidence indicates natural food-derived Gal-3 inhibitors as promising candidates for CRC prevention and therapy. However, critical issues, such as their bioavailability and efficacy, in controlled human studies need to be addressed to translate research progress into clinical applications.

## 1. Introduction

Colorectal cancer (CRC) is the third-most-common cancer and the second-leading cause of cancer-related mortality worldwide, thus, representing a considerable health issue [1,2] (WHO. Cancer. Available online https://www.who.int/news-room/fact-sheets/detail/cancer, accessed on 3 November 2022). The worldwide increase in CRC incidence and mortality has been associated to the overweight/obesity epidemic and to the increasing adoption of Western lifestyles, with a shift in dietary patterns towards a decreased consumption of plant-source foods and an increased intake of fat, sugar and animal-source foods [3]. Epidemiologic research approximates that more than one-half of colon cancer risk is preventable through lifestyle and dietary measures [4,5], and numerous studies confirmed the association between dietary patterns and CRC risk [6,7] (World Cancer Research Fund /American Institute for Cancer Research, https://www.wcrf.org/wp-content/uploads/2021/02/Colorectal-cancer-report.pdf, accessed on 10 October 2022). In particular, prudent dietary patterns and consumption of fruits and vegetables have been inversely correlated with both the risk of developing CRC and CRC-related mortality [8,9]. Strong evidence has also been provided for an inverse association between CRC risk and consumption of fiber-rich foods [6,10,11,12,13,14,15](World Cancer Research Fund/American Institute for Cancer Research, https://www.wcrf.org/wp-content/uploads/2021/02/Colorectal-cancer-report.pdf, accessed on 10 October 2022).

CRC exhibits a great level of complexity, being characterized by several multi-step disease events linked to the accumulation of genetic/epigenetic alterations, as well as high heterogeneity at all disease stages [16]. Dysregulated immune response and immunosuppression in the tumor microenvironment (TME) strongly contribute to tumor growth and invasion. Surgery is the gold-standard treatment for early-stage disease. However, approximately 25% of patients have metastatic disease already at diagnosis and ~50% of CRC-diagnosed patients will develop metastases, mostly in the liver. Even with the recent advances in surgical and therapeutic options, tumor metastases and therapy resistance are the foremost hurdles in CRC management.

Multiple genes are involved in the biology of CRC, including different galectins. Galectins are glycan-binding proteins, whose importance is due to their ability to control the innate and adaptive immune system in health and disease conditions [17,18]. Sixteen galectins, distinguished into three main groups (prototypal, tandem repeat and chimeric) based on the carbohydrate recognition domain (CRD), have so far been identified in mammals [19]. Galectin- (Gal-) 1, -2, -5, -7, -10, -11, -13, -14, -15 and -16 are “prototype” galectins, characterized by the presence of a single CRD, while Gal-4, -6, -8, -9 and -12 belong to the “tandem-repeat” type with two distinct CRDs connected by a short linker peptide [20,21,22,23].

Gal-3 is the only “chimera-type” galectin, whose unique and intriguing structure is composed of a COOH-terminal domain containing one CRD linked to an extended N-terminal (NT) region [20,21,22,23] (Figure 1A). It is ubiquitously produced and can be found in the cytoplasm, in the nucleus or released in the extracellular space [24]. In humans, Gal-3 is encoded by the *LGALS3* gene, located on chromosome 14 locus q21-22 [25], and similarly to several other galectins, is structurally folded as a globule of two antiparallel beta sheets formed by five and six beta strands; the residues involved in carbohydrate binding are part of a pocket formed by four adjacent beta strands [26]. The Gal-3-CRD contains an Asp-Trp-Gly-Arg-motif [27] that is also present in the Bcl-2 family of apoptosis regulators and is responsible for the anti-apoptotic activity of Gal-3 [28]. Its N-terminal region is rich in proline residues (27 in humans), whose functional value is to modulate Gal-3 function [29]. Mutations in each proline within the N-terminal region differentially control NT–CRD interactions, consequently affecting glycan binding, liquid–liquid phase separation and cellular activities [29]. A recent finding has shown that these genetic variants are linked to an increased risk of disorders related to immunity or autoimmunity, such as inflammatory bowel disease (IBD) and type 1 diabetes [30].

The importance of Gal-3 in regulating the immune response and inflammation as well as the transition from acute to chronic inflammation has been amply recognized. However, its involvement in the pathogenesis of intestinal inflammation has not yet been completely clarified [31,32].

Chronic inflammation plays a key role in colon carcinogenesis, and diet has the potential to influence cancer risk and progression by regulating gut inflammation. A relationship among diet and its components and Gal-3 expression and activity has been higlighted. Pro-inflammatory high fat diets (HFDs) and saturated fatty acids increase Gal-3 levels in exposed animals [33,34,35] and Gal-3 overexpression has been reported in diet-related human disorders, such as obesity and type 2 diabetes [36]. Conversely, some molecules (e.g., fibers, polyphenols) in plant-based food, whose antitumor properties have been extensively proved in the last few decades, are gaining importance for their capacity to suppress Gal-3 expression or to act as competitive inhibitors [37].

This review aims to present insights on the multiple relationships between Gal-3 and CRC and to explore how Gal-3 blocking by food-derived bioactive compounds may contribute to cancer prevention and the development of clinical intervention strategies.

## 2. Galectin-3 in Colorectal Cancer Development and Progression

The existence of a coordinated network of galectins exhibiting antagonistic roles in intestinal mucosa inflammation is supported by the literature. By exerting anti- or pro-inflammatory activities, galectin family members affect epithelial barrier integrity and gut immune homeostasis. In particular, Gal-3 has been extensively demonstrated to play a pivotal role in intestinal inflammatory diseases as well as in CRC growth and progression [38]. In this section, we will review the human studies reporting this evidence and summarize the main mechanisms through which Gal-3 regulates the activities of cancer and immune cells during CRC onset and progression.

### 2.1. Role in Human Intestinal Inflammation and Colorectal Cancer

Gal-3 has been reported to play both pro- and anti-inflammatory roles, depending on the body district and subcellular localization [39]. In the context of intestinal mucosa, Gal-3 shows mostly pro-inflammatory functions, even though a regulatory role, limiting the inflammatory process and restoring mucosal homeostasis, has also been described in IBD [32]. This can be explained by its dual role in protecting T cells from apoptosis when present intracellularly [27], while promoting apoptosis when acting from the extracellular space [40,41].

Gal-3 is constitutively expressed within the epithelial compartment of the intestine in both mice [42] and humans [43]. In humans, Gal-3 is expressed in the normal colon in nuclear compartments (Figure 2A), but significant changes in its expression and/or subcellular localization have been observed both in IBD and CRC [38]. As depicted in Figure 2B, a progressive increase in Gal-3 expression is observed in adenoma and during the progression toward advanced cancer, with changes in its subcellular localization from the nucleus in adenomas to cytoplasm in CRC.

Accordingly, serum levels of Gal-3 are significantly elevated in both ulcerative colitis (UC) and Crohn’s disease (CD) patients compared to healthy people [44,45]. Furthermore, enhanced Gal-3 concentration in serum and stool samples of patients with UC correlates with clinically and histologically severe disease [46], thus, representing a valuable biomarker for monitoring UC progression. The results achieved in UC patients are in keeping with data obtained in a murine model of dextran sodium sulphate (DSS)-induced colitis, where Gal-3 expression promotes acute colitis and plays an important pro-inflammatory role in the colonic epithelium [47]. However, the potential use of serum Gal-3 levels as an IBD biomarker is still controversial since recent studies failed to show increased serum levels of Gal-3 in UC and CD patients [48]. Differences in factors that affect the levels of Gal-3, such as age, renal function and co-morbidities, as well as the stage of the disease, could explain the discrepancies observed within IBD studies. Moreover, higher serum and fecal content of Gal-3 was observed in subjects with UC and metabolic syndrome, showing clinically and histologically milder disease compared to subjects suffering from UC only [49], indicating that Gal-3 may also be involved in the mechanism limiting the inflammatory process in UC. Notably, IBD is associated with an increased risk of CRC that appears to raise with the duration, severity and extent of colonic inflammation [50].

Increased Gal-3 expression has been amply recognized in colon carcinogenesis [38,51,52] and is considered a prognostic factor of poor outcome [44]. An interesting study by Tao and colleagues, examining Gal-3 expression in CRC cases in both cancer tissue and adjacent normal tissue via immunohistochemistry, revealed an association between Gal-3 levels and the clinical/pathological characteristics of disease [38]. However, conflicting results emerged on the relationship between the pattern of Gal-3 expression and tumor progression because some studies reported decreased or comparable Gal-3 levels during CRC progression [53,54,55,56,57,58,59]. A possible explanation for this discrepancy comes from the study of Tsuboi and co-workers, who, focusing on Gal-3 expression changes in two different tumor areas, highlighted the presence of a significant number of liver metastases when the expression of Gal-3 was lower at the invasive front of the tumor as compared to its surface [52]. Additionally, emerging evidence reveals that patients with detectable expression of Gal-3 in the tumor have more lymph node and distant metastases, frequent venous invasion and deeper wall invasion in comparison to Gal-3-negative cases [51]. Further, in clinical settings, Gal-3 levels were found to be higher also in feces from patients with severe tumor stage, suggesting the utility of this lectin as a possible biomarker for disease severity and progression [49]. Likewise, the potential use of Gal-3 as a predictive marker of therapy response also emerged from proteomics analysis studies carried out in rectal cancer patients showing Gal-3 protein downregulation in rectal tissues after radiation treatment [60].

Gal-3 is also released into the circulation and a strong expression (up to 5-fold increase) has been reported in the bloodstream of CRC patients [61]. Particularly, circulating Gal-3 has been found to induce the secretion of cytokines contributing to tumor progression, thus, suggesting that Gal-3 could function as a pro-inflammatory mediator during the metastatic cascade [62,63]. Another interesting study evaluated the influence of Gal-3 on postoperative outcomes after gastrointestinal surgery. As shown by Matsuda and colleagues, the high levels of Gal-3 found in the perioperative blood, likely transiently induced by surgical stress, are associated with postoperative complications (POCs) following surgery [64]. Therefore, while, on the one hand, it is unlikely that patients with blood Gal-3 levels below the cut-off value develop POCs, on the other hand, those with higher Gal-3 levels can be considered at risk of POC. Deep insights regarding the clinical value of circulating Gal-3 for young CRC patients have also been reported. It has been recently found that the combined detection of serum Gal-3, Aquaporin-1 (AQP-1) and -3 (AQP-3) (integral membrane proteins that serve as passive channels for water) has potential value for the diagnosis of young patients with colon cancer [65].Specifically, in young patients with colon cancer, pre- and postoperative Gal-3, AQP-1 and AQP-3 serum concentrations were found to be significantly increased, thus demonstrating their importance in evaluating long-term prognosis [66]. A multi-center clinical study was recently undertaken in 13000 asymptomatic individuals to validate Gal-3 and other factors as biomarkers for the early detection of colon cancer [67]. Moreover, clinical research has recently begun to explore the role of Gal-3 in treating cancer and intervention with Gal-3 antagonists is emerging as an attractive option for CRC. A few phase I or II human trials are ongoing in advanced CRC patients with the aim of assessing the safety and efficacy of these inhibitors in combination with standard therapies [67].

### 2.2. Direct Effects on Tumor Cells

Dysregulated expression of Gal-3 and changes in its localization are involved in regulating multiple processes. Gal-3 synthesis occurs in the cell cytoplasm [24], from where it can be translocated into the nucleus [68] or alternatively secreted into the extracellular space [69]. Changes in its localization (intra- or extracellular) determine its numerous functions. In the intracellular compartment, Gal-3 exhibits dynamic behavior, regulating signal transduction pathways [70], exerting anti-apoptotic activity [71] and contributing to pre-mRNA splicing [72]. In the extracellular space, Gal-3 regulates cell adhesion and is implicated in the organization of the plasma membrane and modulation of tumor invasion and metastasis [73,74]. Other representative examples of extracellular effects include the regulation of immune surveillance [75] and glycoprotein endocytosis [76].

The effect of Gal-3 on apoptosis constitutes the main difference between intracellular and extracellular Gal-3. The intracellular one is anti-apoptotic and is involved in cell proliferation and differentiation, while the extracellular Gal-3 activates inflammatory Th1 and Th17 cells and is involved in target cell apoptosis. Once secreted, Gal-3 interacts with the extracellular matrix and cell surface glycans. In the presence of carbohydrate ligands, it can oligomerize (Figure 1B), forming lectin lattices that act as scaffolds, sustaining the spatial organization of cell surface signaling receptors (e.g., epidermal growth factor receptor, platelet-derived growth factor receptor, fibroblast growth receptor, vascular endothelial growth factor receptor and transforming factor-β receptor) [77,78]. This complex interaction favors the survival of tumor cells in stressed conditions, induces tumor cell detachment and migration, and attracts leukocytes and endothelial cells to the tumor environment, thus, helping angiogenesis [78].

Regardless of its localization, increased levels of Gal-3 are related to increased CRC risk and severity. Particularly, it has been demonstrated that Gal-3 overexpression increases the migration of colon cancer cells through the activation of the K-Ras–Raf–Erk1/2 pathway [79] and that its interaction with extracellular carcinoembryonic antigen (CEA) promotes the migration of cancer cells and the appearance of distal metastases. In addition, synergistic interaction between serum Gal-3 and CEA correlates with poor survival in CRC patients [80]. Wu and co-workers investigated the effect of extracellular Gal-3 on colon cancer cell migration and its correlation with EGFR expression. Their results showed that extracellular Gal-3 increases cancer cell migration, which correlates with EGFR levels. These results suggest that Gal-3 targeting may have a combined effect on EGFR-targeted therapy [81].

Furthermore, by interacting with the oncofetal Thomsen–Friedenreich carbohydrate antigen (Galβ1, 3GalNAcα-, TF) on cancer-associated transmembrane mucin protein 1 (MUC1), circulating Gal-3 induces MUC1 cell surface polarization and exposure of cell adhesion molecules. This leads to the formation of emboli, prolonging survival of disseminating tumor cells in the circulation, thus, favoring the metastatic cascade [82]. Through its direct interaction with MUC1, Gal-3 may activate the MAPK and PI3K/Akt signaling pathways, leading to enhancement in cell proliferation and motility on the cell surface. This Gal-3-triggered MUC1-mediated signaling promotes uncontrolled tumor cell malignancy [83].

Growing evidence indicates that the Wnt/β-catenin pathway is implicated in the maintenance of cancer stem cells in CRC. This finding provided the rationale to investigate the role of Gal-3, known to be involved in Wnt signaling, on the Wnt/beta-catenin pathway in colon cancer cells. The results indicate that Gal-3 regulates β-catenin expression and its nuclear accumulation and activates Wnt signaling by regulating GSK-3β activity via the PI3K/AKT pathway [84,85].

An additional aspect worth deeper investigation is Gal-3’s role in chemo-sensitivity, invasion and metastasis in colon cancer. The data reported by Lu and colleagues unraveled the existence of an miR-128/Gal-3 axis in CRC. In particular, a frequent miR-128 down-regulation, negatively correlating with Gal-3 level, has been shown in CRC. A decreased expression of miR-128 was associated with CRC progression and a worse patient prognosis [86]. The shift in Gal-3 localization is also crucial in the response to chemotherapeutic drugs. Through a complex mechanism, including five steps, Gal-3 inhibits apoptosis induced by anticancer drugs. After a process of phosphorylation and translocation from the nucleus to the cytoplasm in response to chemotherapeutic drugs, Gal-3 activates a pathway that promotes the stabilization of the mitochondrial membrane; this results in the suppression of cytochrome c release and caspase activation, thus, favoring the suppression of cell apoptosis [40].

Gal-3 is also involved in a resistance mechanism of colon cancer cells to TRAIL-induced cell death. TRAIL can induce apoptosis and preferentially kills tumor cells by engaging specific death receptors, i.e., DR4 and DR5. However, tumor cells may resist TRAIL-based therapy in many ways. Findings from Mazurek and co-workers highlighted a mechanism by which Gal-3 inhibits trafficking of death receptors by anchoring them in glycan nano-clusters, thus, blocking the apoptotic signaling execution [87].

### 2.3. Contribution to Colonic Inflammation and Immunosuppression

In addition to the direct effects on colon cancer cells, Gal-3 regulates immune cell function in both innate and adaptive responses, where it participates in the activation or differentiation of many immunocompetent and inflammatory cells, including neutrophils, monocytes/macrophages, eosinophils, mast and dendritic cells (DCs) and activated T and B cells. Gal-3 interaction with immune cells inhibits the normal functions of the immune system, potentially mediating the immune escape of tumor cells and promoting tumor-driven immunosuppression within the TME, through several mechanisms [88]. In general, Gal-3 has been widely studied in the context of acute inflammatory responses [89] where it provides a powerful pro-inflammatory signal. Through intracellular or extracellular mechanisms, this lectin controls inflammatory responses by modulating cell adhesion, migration, function and survival of various innate immunity cells. The specific effects of Gal-3 on the innate and adaptive immunity are extensively reviewed elsewhere [39,90].

Regarding to intestinal chronic inflammatory conditions and colon carcinogenesis, Gal-3 expression showed different effects on the function of both colon-infiltrated macrophages and T cells, the most important effector immune cells involved in the progression of colon inflammation, as summarized in Table 1.

Specifically, Gal-3 is highly expressed in colonic macrophages in UC patients. In mouse models of experimental colitis, its deficiency inhibits the activation of the NLRP3 inflammasome and the production of inflammatory cytokines in colonic macrophages as well as in mesenteric lymph nodes, resulting in an attenuation of disease severity [47,93]. Likewise, pharmacological inhibition of mesenchymal stem cell (MSC)-produced Gal-3 enhances the capacity of these cells to promote alternative activation of peritoneal macrophages, both in vitro and in vivo [91]. Altogether, these findings indicate that Gal-3 inhibition could be used for improvements in MSC-mediated polarization towards the immunosuppressive M2 phenotype of macrophages in UC.

Conversely, in humans, Muller and colleagues found reduced Gal-3 expression in the inflamed mucosa of CD and UC patients and demonstrated that the inhibition of Gal-3 deeply influences T-cell activity [98]. In line with this evidence, after treatment with recombinant Gal-3, T cells in UC patients developed an immunosuppressive phenotype and were not able to optimally proliferate [92]. Additionally, adoptive transfer of Gal-3-primed T cells significantly reduced bowel inflammation and disease severity by inducing regulatory T cells, which suppressed colon mucosal inflammation [92]. Finally, a recent study by Volarevic and colleagues showed, in a chronic DSS-induced colitis model, that Gal-3 regulates the immunosuppressive capacity of DCs in the gut via a TLR-2-dependent activation of the IDO-1/KYN pathway, promoting the expansion of colon-infiltrated T-regulatory cells, which, consequently, suppress Th1 and Th17 cell-driven colon inflammation [99].

With respect to CRC, several studies have demonstrated that a high level of expression of Gal-3 promotes tumor growth, both in vivo and in vitro. Using a mouse tumor model, Peng and co-workers [94] demonstrated that Gal-3 treatment at high doses abrogates the efficacy of tumor-reactive T cells, induces T-cell apoptosis and promotes tumor immune tolerance. Additionally, Gal-3 null mice showed delayed T-cell, macrophage and DC infiltration into the gut mucosa after inoculation with *Citrobacterrodentium*, together with a slight delay in the resolution of infection [95]. Gal-3 serum concentration was also examined in patients with untreated CRC in association with interleukin (IL-10, IL-12 and IL-17) production, lymphocyte activation and inflammatory parameters, such as neutrophil/lymphocyte ratio (NLR), white blood cell count (WBC) and C-reactive protein (CRP) [96]. The levels of circulating Gal-3 showed a significant positive correlation with IL-17 and IL-10 production, whereas an inverse correlation was observed with IL-12. Thus, the effect of Gal-3 on the production of effector Th1-promoting cytokines may depress anti-tumor cell-mediated immunity, through Th2-dominant conditions. On the other hand, higher Gal-3 levels were associated with higher inflammatory parameters (NLR, WBC, CRP) and lower lymphocyte stimulation [96], according to the role of chronic systemic inflammation in the suppression of tumor immunity.

In vitro, Gal-3 inhibits the expansion of T cells, promotes CD8 T cell apoptosis, alters macrophage polarization from M1 (anti-tumor) to M2 (pro-tumor) and limits TCR clustering, once secreted by tumor cells [100]. Among the molecular mechanisms accounting for the powerful lymphocyte inhibitory effect of Gal-3 in cancers, this protein modulates the interactions between T cells and antigen-presenting cells, thus, playing a central role in the initial steps of tumor antigen presentation [100]. Specifically, the conditioned medium derived from colon cancer cell lines significantly induces the expression of Gal-3 in THP-1 monocytes and actively influences the phenotype of monocytes, switching their differentiation into a population of non-adherent mixed M1 and M2 cells [97]. These results suggested that colon cancer cells influence monocyte differentiation into suppressive subsets, likely via Gal-3 production. Interestingly, macrophage supernatants modulate the expression and secretion of Gal-3 in colon cancer cells [101], indicating the existence of a bidirectional cross-talk between tumor and immune cells.

Finally, along with the pro-inflammatory and immunoregulatory functions, Gal-3 also exhibits anti-microbial activities. In the small intestine, its increased expression in epithelial cells and macrophages can affect microbiota composition and macrophage activation [102]. Moreover, Gal-3 may interact with commensal bacteria, possibly influencing their colonization capacity [103]. Despite Gal-3 abundance at gastrointestinal sites, little is known yet about its interaction with the microbiota.

## 3. Galectin-3 Targeting by Bioactive Food Compounds in Colorectal Cancer Prevention and Therapy

Given the prominent role of extra- and intracellular Gal-3 in vital processes of carcinogenesis, the development of efficient Gal-3 inhibitors gained attention in the field of cancer prevention and therapy. Those currently synthetized are predominantly based on competitive blocking of the interaction between Gal-3 CRD and natural carbohydrate binding partners on the cell surface or inside the cells. The most outstanding inhibitors that have moved to the stage of preclinical testing are heterogeneous polysaccharide-based molecules [104]. At the same time, research interest is devoted to the study of natural diet-derived complex carbohydrates or other compounds to be exploited as Gal-3 inhibitors. Indeed, some bioactive food compounds, mostly derived from fruits and vegetables, have been explored for their capacity to negatively regulate Gal-3 expression or to act as competitive inhibitors by targeting the Gal-3 CRD–glycan interaction [105].

In this section, we summarize the studies that have highlighted the role of non-digestible carbohydrates and polyphenols in controlling CRC development or growth through the regulation of Gal-3 activity/expression. Their widespread anti-Gal-3 effects and their role as promising tools for cancer therapy and prevention are discussed.

### 3.1. Non-Digestible Carbohydrates

Dietary fibers are a heterogeneous group of plant-derived complex carbohydrates that are indigestible for the host and fermented by gut bacteria, the regular consumptionof which supports colonic health and prevents local carcinogenesis [106,107]. Non-digestible carbohydrates (NDCs) consist of complex carbohydrates from the plant cell wall, including pectin and hemicellulose, which vary among foods in structure and composition and, consequently, in physicochemical properties and biological effects. Water-soluble fibers are mainly composed of pectin, which is considered a bioactive polysaccharide due to its beneficial effects on human health and has emerged as a good source for generating high-affinity Gal-3 inhibitors with low toxicity [108]. In addition to being a major component in fruits, vegetables and whole grains, extracted pectin is also used in the food industry as a gelling, thickening, emulsifying and stabilizing agent in a variety of foods [109]. The food source, pectin extraction methods and ripening parameters of fruits are important factors in obtaining functional pectin molecules. Several fruits (apple, plum, apricot, jaboticaba, citrus, lemon and papaya) as well as ginseng, pumpkin, olive, tomato and corn, have been explored for pectin extraction and biological activity studies.

NDCs have the potential to affect cancer risk through different mechanisms, including the influence on the gut microbiota and the regulation of immune responses [110]. By virtue of their capacity to bind the CRD of Gal-3, they interfere with its attachment to glycan-containing surfaces [111] and can inhibit the Gal-3 downstream signaling mechanisms that controls apoptosis and cell cycle in CRC cells as well as the activation of tumor-specific immune responses [112]. Several in vitro and in vivo studies have investigated the interaction between distinct NDC and Gal-3, as well as the therapeutic and chemopreventive effects of this interaction on CRC [37,113] (Table 2).

The modified citrus pectin (MCP) has attracted special attention, being the most studied among NDCs. MCP is a preparation derived from the white pith of citrus fruit peels that is modified by high temperature, alterations in pH and/or pectinase treatment, resulting in smaller and less-ramified galactan chains, which have better access to Gal-3 CRD [126]. MCP effectively binds to recombinant Gal-3 and inhibits Gal-3-mediated functions, such as homotypic tumor cell aggregation, binding of tumor to endothelial cells, anchorage-independent growth, angiogenesis and tumor-infiltrating lymphocyte impairment [125].

Early studies reported the beneficial effects of MCP dietary supplementation in animal models of colon cancer [127]. Nangia-Makker and co-authors reported a strong therapeutic effect of oral MCP intake, with inhibition of both primary tumor and liver metastases (LM) growth [114]. MCP was found to reduce tumor growth in vivo by acting as an inhibitor of angiogenesis. In parallel with in vitro assays, it prevented HUVEC migration and capillary tube formation by binding to Gal-3 in the matrix and on the endothelial cells and interfering with receptor recognition. This mechanism was assumed to be responsible for in vivo therapeutic efficacy [114,128]. Accordingly, a dose-dependent inhibitory effect on LM was reported in colon-cancer-bearing mice following oral MCP administration [115]. The therapeutic effect correlated to the high levels of Gal-3 expression found in metastatic lesions, arguing for the blocking of Gal-3 function as the main mechanism of MCP-induced protection [115]. MCP-mediated control of tumor growth was also reported in nude mice engrafted with SW-480 cells where this compound reduced Gal-3 expression and induced apoptosis and reversion of epithelial–mesenchymal transition (EMT) in the engrafted tissue [116].

The chemopreventive activity of alginate polysaccharide from citrus, combined with *Lactobacillus acidophilus*, was studied in a mouse model of AOM-induced colon carcinogenesis. Alginate MCP dramatically reduced the occurrence of precancerous lesions through Gal-3 and VEGF blocking [117]. Furthermore, oral intake of a modified polysaccharide extracted from apple, mainly including Gal-3 binding galacturonic acid and galactose, protected mice against DMH/DSS-induced colon tumorigenesis [118]. Finally, a recent study demonstrated that SCLP, a pectin-type polysaccharide purified from Smilax china Linn (or China roof), can significantly improve inflammation and the histopathological damage of DSS-induced UC in mice by blocking the interaction of Gal-3 with the NLRP3 inflammasome [3]. The Gal-3-mediated anti-inflammatory activity of this pectin could play a role in CRC prevention, with UC-associated chronic inflammation being a risk factor for colon carcinogenesis.

The antitumor effects of MCP and pectin from other sources have also been explored in in vitro studies. A study by Wu and colleagues showed that extracellular Gal-3 is involved in lamellipodia formation and migration of colon cancer cells and that MCP is able to significantly reduce this effect [81]. Other studies have found that MCP inhibits cancer cell proliferation and aggregation and induces EMT reversion through reductions in Gal-3 interaction with cells and between cells and the extracellular matrix [111,116].

Modified sugar beet, papaya and ginseng pectins have structures composed of neutral (1,4)-β-galactose residues, which were related to Gal-3 interaction and inhibition. As observed for MCP, the binding of these pectins to Gal-3 is associated with strong effects on colon cancer cell proliferation and migration, particularly evident following temperature-, alkali- or enzyme-induced modifications [119,122]. Furthermore, structural studies revealed that the rhamnogalacturonan I-rich fragment RG-I-4 from ginseng pectin alone holds the capacity to inhibit Gal-3-induced hemagglutination and binding to T lymphocytes, and to block colon cancer cell adhesion and aggregation [120]. Likewise, the use of specific enzymes, able to increase the proportion of RG-I regions on apple pectin, equips this molecule with a superior ability to induce apoptosis and ROS production in colon cancer cell lines and to enhance the cytotoxic activity of anticancer drugs [121]. More recently, pectin-rich olive extracts from the byproduct of olive oil production have been tested for their antitumor activity in comparison with MCP. Olive-derived pectin was found to inhibit Gal-3 activity and to reduce colon carcinoma cell proliferation with higher efficiency than MCP [123].

Some fruit flours, consumed as regular flour or incorporated in products, represent an alternative for fiber consumption. The pectin-rich water-soluble fractions of commercial plum, papaya and jaboticaba fruit flours were studied for their inhibitory effect on Gal-3 and for their potential capacity to affect colon cancer cell growth [124]. Only pectin extracted from jaboticaba fruit flour significantly inhibited Gal-3-mediated hemagglutination and decreased the viability of Gal-3 overexpressing HCT116 cells [124]. Notably, the depolymerization of polysaccharides, occurring during heating to produce fruit flours, decreases their beneficial health effects. In keeping with this observation, when pectin extracted from the papaya pulp was tested, inhibition of both Gal-3 activity and colon cancer cell growth was obtained [108,125,129]. The inhibitory effect was also influenced by variations in polysaccharide composition naturally occurring during fruit ripening and was particularly evident at early/intermediate ripening stages [108,110,125], suggesting that the consumption of papayas during intermediate ripening time points could be promising for CRC prevention.

A number of preclinical studies have analyzed the Gal-3-blocking effects of pectins, isolated from different fruits, vegetables and cereals, in other cancer models (melanoma, prostate, breast, bladder, thyroid), as well as in cardiovascular diseases, with promising results. The exceptionally low incidence of toxicities and possible clinical effects of pectin, qualified by the FDA as a GRAS (i.e., generally recognized as safe) food substance, support further testing of these dietary compounds in human studies [113]. Indeed, although studies involving CRC patients are lacking [130], a number of clinical trials are ongoing to assess the therapeutic or preventive value of pectin in cancer and other diseases [131], some with promising results [132,133]. Specifically, in a prospective phase II study using MCP (PectaSol^®^, P-MCP) in non-metastatic biochemically relapsed prostate cancer patients, a significantly reduced disease progression was observed [132]. Likewise, the use of Belapectin, in association with immune checkpoint inhibitor-based therapy, in a phase I trial involving patients with advanced melanoma or head and neck squamous cell carcinoma, resulted in enhanced anti-tumor immune response [133]. Notably, MCP was approved as a dietary supplement with antioxidant and immunomodulatory properties.

### 3.2. Polyphenols

Polyphenols are bioactive food compounds mainly found in fruits, vegetables, whole grains and plant-derived beverages (fruit juice, red wine, tea and coffee) [134]. They exert strong anti-inflammatory, antioxidant and immunomodulatory activities, and growing evidence from epidemiological studies has suggested a possible inverse association between polyphenol intake and cancer incidence/mortality [135]. Polyphenols are the most-studied dietary compounds in the field of oncology and have received great attention in CRC research for both chemopreventive and direct anticancer activities, and for their novel role as enhancers of conventional treatments [136,137]. A wide range of clinical trials have been registered, in which some polyphenolic compounds (e.g., curcumin, genistein, fisetin, silybin, epigallocatechin gallate) are being tested as preventive and anti-cancer medicines, mostly in combination with other drugs, in different cancer types, including CRC and precancerous colon adenomatous polyposis [136,138,139].

Some recent studies have highlighted the capacity of these compounds to regulate Gal-3 expression or activity in experimental models of cancer and other pathologies. The results achieved suggest that their capacity to exert inhibitory effects on Gal-3 contributes to their antitumoral activity (Table 3).

Among polyphenols, resveratrol, a compound attracting considerable interest during the last few decades as an antitumor agent, was found to downregulate Gal-3 expression in ovarian cancer cells through the induction of mir-424-3p [140]. Gal-3 blocking resulted in the inhibition of cancer cell migration and invasion, and in their increased sensitivity to chemotherapy [140]. In a different experimental disease model, HFD-induced atherosclerosis, Li H and colleagues demonstrated that the flavonoid quercetin alleviates atherosclerotic lesions by inhibiting HFD-induced Gal-3 expression and consequent NLRP3 inflammasome activation in macrophages [141]. The negative effect of quercetin on the Gal-3-NLRP3 pathway was also confirmed in in vitro stimulated macrophage cell lines [141]. Accordingly, treatment with berberine suppressed oxidized low-density lipoprotein-induced upregulation of Gal-3 and macrophage activation [142]. Quercetin and berberine were both shown to exert beneficial effects on intestinal inflammation and to inhibit colon tumorigenesis [144,145]. Finally, the effects of xanthohumol and 8-prenylnaringenin, belonging to the prenylflavonoid family, on Gal-3 expression were investigated in a mouse model of type 2 diabetes mellitus [143]. HFD-induced diabetes resulted in overexpression of Gal-3 in the liver and kidney that was associated with oxidative stress. Oral intake of both polyphenols completely suppressed Gal-3 expression and oxidative stress markers [143]. Of note, these compounds have been reported to affect mitochondrial function and oxidative pathways in CRC [146].

Although the effects of polyphenols in regulating Gal-3 activity have been poorly investigated in cancer, the results achieved in different disease models point to Gal-3 inhibition as a novel mechanism underlying the anti-inflammatory and anti-tumoral activity of these compounds. Polyphenol-mediated effects on Gal-3 have not yet been specifically explored in CRC models, but the observation that Gal-3 is highly expressed in CRC tissues, together with its known role in intestinal inflammation and colon carcinogenesis, suggest that the capacity of these compounds to modulate Gal-3 in other disease contexts might also be exploited in the management of CRC, thus, further strengthening the potential of their supplementation for therapeutic and prevention purposes.

## 4. Conclusions

The mechanisms underlying CRC onset and progression are numerous and complex. Despite the great efforts devoted to the management of CRC, it still represents a leading cause of death worldwide. Although a large choice of targeted treatments is available, more individualized therapies need to be developed to achieve longer survival, fewer adverse reactions and the potential for full recovery.

In this review, we summarized the multiple tumor-promoting and -supporting roles played by Gal-3, from the contrasting effects on apoptosis to TME immunosuppression, tumor angiogenesis and metastasis promotion. Worthy of note, a growing body of evidence highlights the importance of Gal-3–ligand interaction in the pathological processes of colon inflammation and CRC development. Likewise, the immunomodulatory effects of Gal-3, indirectly affecting tumor development/progression, are particularly relevant for CRC and other gastrointestinal cancers whose onset and evolution are deeply influenced by the immune system. Furthermore, the observation that Gal-3 overexpression is significantly related to tumor progression points to this lectin as a potential candidate to predict CRC prognosis. The clinical data currently available, however, are still insufficient to clearly validate the usefulness of Gal-3 as a prognostic biomarker of CRC.

Finally, we focused our attention on the preventive/therapeutic potential of Gal-3 targeting natural food compounds. It is very likely that new benefits will be discovered for these compounds as Gal-3 research continues to identify novel mechanisms underlying Gal-3-mediated disease induction/progression. This will pave the road towards more targeted hypotheses for human study interventions and clinical trials. While surgery and chemo- and radiotherapy interventions continue to represent essential treatments in CRC management, depending on the tumor stage, the experimental evidence reported herein strongly indicates Gal-3 targeting by natural food compounds as a potential intervention strategy, with important implications in the therapeutic management of CRC patients.

## 5. Future Directions

Fibers and polyphenols exert a broad-spectrum power as anticancer and health-promoting agents. Their Gal-3-independent regulatory effects on inflammation, oxidative stress and microbiota composition have been extensively proved in human studies. The anti-metastatic effects of these compounds associated with Gal-3 blocking hold great promise for the control of CRC and other tumors, given the impact of metastases on therapy efficacy and postoperative prognosis. Thus, more research is needed to definitely prove the connection between their intake and in vivo blocking of Gal-3 activities. The contribution of Gal-3 blocking to the anti-tumoral potential of bioactive food compounds in CRC should be more deeply investigated and considered in future intervention studies. A schematic model on the preventive/therapeutic potential of Gal-3 targeting by these compounds is depicted in Figure 3.

Furthermore, as the effects of these molecules can vary widely according to their structure, administration route and in vivo bioavailability, additional structural composition studies are recommended. Likewise, more attention needs to be paid to still unclear aspects accounting for their systemic and tumor bioavailability following food consumption. Consequently, the recommendations for their intake are expected to change from a quantitative to a qualitative perspective, thus, improving nutritional intake for both healthy individuals and cancer patients.

The exploitation of healthy diets or food-derived bioactive compounds able to target multiple tumor-promoting/supporting pathways represents a desirable goal in cancer prevention and a strategy to enhance the efficacy of conventional therapies. An additional aspect concerns ecological sustainability. As many of the possible sources of pectin, polyphenols and other bioactive molecules are residues and byproducts from agriculture and food-processing industries, their exploitation in clinical studies could lead to a revalorization of materials otherwise discarded.

## Figures and Tables

**Figure 1 cancers-15-00052-f001:**
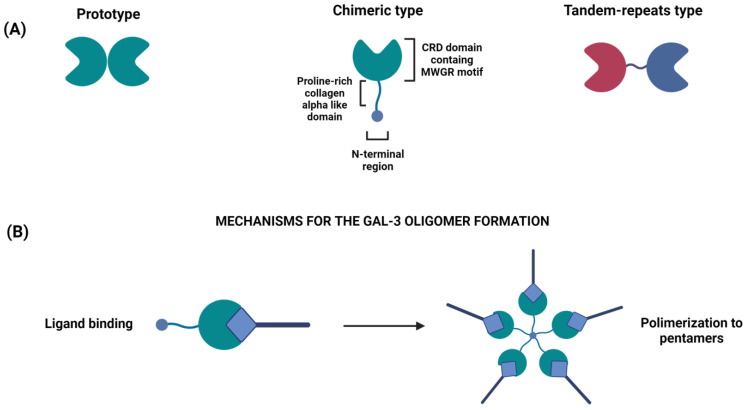
(**A**) Classification of the different types of galectins: prototype (Gal-1, -2, -5,-7,- 20,-11, -13, -14, -15 and -16), chimeric type (Gal-3), tandem-repeat type (Gal-4, -6,-8,-9 and -12). Prototype galectins have one CRD that can dimerize; tandem-repeat galectins have two different CRDs (red and blue colors are used to distinguish them) linked by a linker domain; chimera-type galectin (Gal-3) has a single C-terminal CRD and a N-terminal oligomerization domain. (**B**) Gal-3 oligomerization: upon oligosaccharide binding by the C-terminal domain, the non-lectin domains of Gal-3 oligomerize, thereby cross-linking ligands of the cell surface.

**Figure 2 cancers-15-00052-f002:**
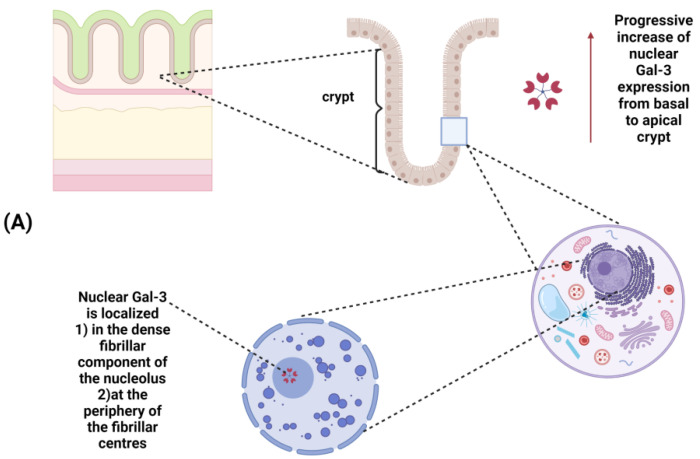
(**A**) Galectin-3 expression in normal colonic mucosa: a progressive increase in Gal-3 expression occurs from the basal to the apical side of the crypt; within the nucleus, Gal-3 is localized in the dense fibrillar components of the nucleolus as well as in the periphery of the fibrillar centers.(**B**) Changes in Gal-3 expression and/or subcellular localization during colorectal cancer progression: Gal-3 expression is increased in advanced cancer. A different subcellular localization (from the nucleus to the cytoplasm) is observed during progression from colorectal adenoma to carcinoma.

**Figure 3 cancers-15-00052-f003:**
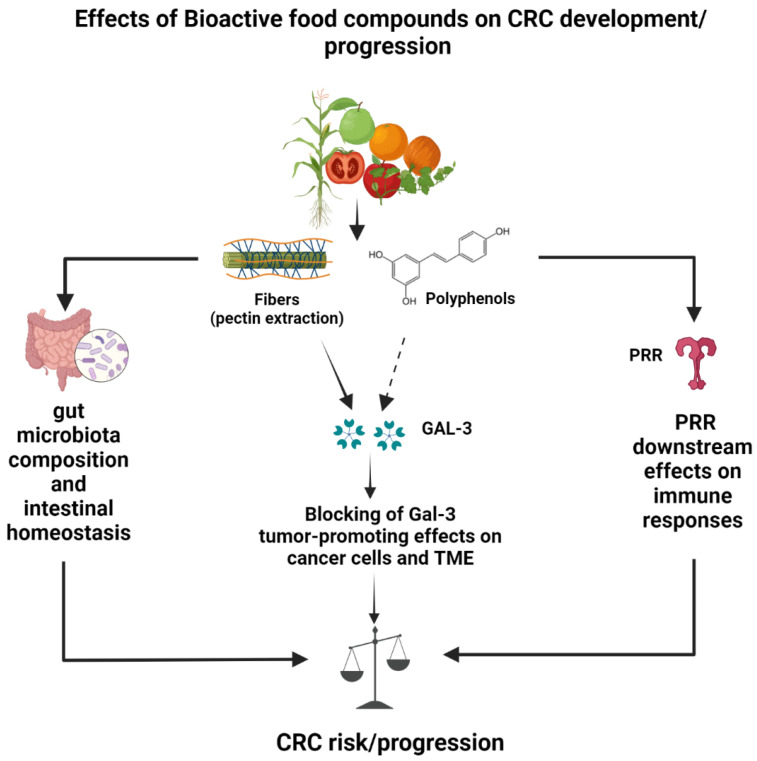
Schematic model of galectin-3-dependent and -independent effects of dietary fibers and polyphenols potentially leading to reduced CRC risk and progression. Galectin-3-independent protective effects occur through modulation of gut microbiota and attenuation of inflammatory responses induced by pattern recognition receptor (PRR) triggering in immune cells. Galectin-3-dependent protective effects on CRC have been demonstrated for pectin (full arrow) and, so far, only suggested for polyphenols (dotted arrow).

**Table 1 cancers-15-00052-t001:** Studies describing the effects of galectin-3 on the immune system in the context of ulcerative colitis and colorectal cancer.

Disease	Model	Immune Cell Type	Investigated Endpoints	Major Findings	Reference
UC	Human (10 patients)	Colonic macrophages	↑Gal-3 expression	Attenuation of acute colitis	[47]
		PBMCs	↑Gal-3 secretion		
	Mice (DSS- induced acute colitis)	↓ Macrophages in colon	Gal-3 blocking (Gal-3^−/−^ ko mice or pharmacological inhibition): ↓NLRP3 inflammasome ↓inflammatory cytokines (IL-1β, TNF-α) ↑IL-10 producing M2 phenotype		
		↓DCs in colon			
		↓ Neutrophils in colon			
UC	Mice (DSS- induced acute colitis)	Colonic macrophages	Gal-3 blocking in MSC (pharmacological inhibition): ↑ immunosuppressive M2 phenotype ↑IL-10	Attenuation of severity of colitis	[91]
UC	Mice (DSS- induced colitis)	T cells	Gal-3 blocking (Gal-3^−/−^ ko mice): ↑ severe disease activity Recombinant Gal-3: ↑ Treg cell phenotype (FOXP3, ICOS, and PD-1 positive) Adoptive transfer of Gal-3 treated T cell: ↓Inhibition of colonic mucosa inflammation	Reduction of disease severity by inducing Treg	[92]
UC	Mice (DSS- induced colitis)	Mesenteric lymphnodes		- Attenuation of acute colitis - Reduction of coloninflammation	[93]
UC	Mice (DSS- induced colitis)	Macrophages	Gal-3 blocking (Gal-3^−/−^ ko mice): ↓ IL-1-, TNF-α, IL-6 and IL-12-producing M1 phenotype ↑ IL-10-, IL-4, and TGF-β- producing M2 phenotype	Reduction of colon inflammation	[46]
		DCs	Gal-3 blocking in DCs (pharmacological inhibition): ↑ DCs immunosuppressive function		
		T cells	↑ colon-infiltrated Tregs ↑pro-inflammatory Th1/Th17 phenotype		
CRC	In vitro, colon cancer cell lines from CRC patients	T cells	Recombinant Gal-3: ↓ tumor-reactive T cells ↑ cell apoptosis	Promotion of tumor immune tolerance	[94]
CRC	Mice (Gal-3^−/−^ ko)	↓ Macrophages in colon	↑IL-6, KC	- Delayed immune cell infiltration into the gut mucosa - Delayed clearance of pathogens	[95]
		↓ DCs in colon			
		↓ T cells in colon			
		= Neutrophils in colon			
CRC	Human (50 patients)	PBMCs	↑Gal-3 secretion ↑ IL-17 and IL-10 ↓ IL-12 ↑inflammatory parameters (NLR, WBC, CRP) ↓limphocytestimulation	- Th2-dominant conditions - Inhibition of cell-mediated immunity	[96]
CRC	In vitro, colon cancer cell lines (HT29, LS180, SW948, SW620)	Monocytes (THP1 cells)	↑Gal-3 and IDO induction by tumor CM	Induction of immune suppressive macrophages	[97]

Abbreviations: CM, conditioned medium; CRC, colorectal cancer; CRP, C-reactive protein; DCs, dendritic cells; DSS, dextran sulphate sodium; IBD, inflammatory bowel disease; IDO, indoleamine 2,3-dioxygenase; KC, keratinocyte-derived chemokine; ko, knock out; NLR, neutrophil/lymphocyte ratio; PBMC, peripheral blood mononuclear cells; siRNA, short interfering RNA; Treg, regulatory T cells; UC, ulcerative colitis; WBC, white blood cell count.

**Table 2 cancers-15-00052-t002:** Studies describing galectin-3-mediated effects of non-digestible carbohydrates in colorectal cancer.

PectinSource and Modification	In Vivo Model	In Vitro Model	Galectin-3 Blocking	Main Results	Reference
Citrus (pH and temperature-modified pectin) 1% in drinking water	NCR nu/nu mice injected with LSLiM6 cells	-	↓ binding to HUVEC ↓ HUVEC chemotaxis/ capillary tube formation	Therapeutic effect: ↓ tumor growth ↓tumor- associated blood vessels ↓ NM and LM	[114]
Citrus (modified pectin) 1–5% in drinking water	Balb/c mice spleen injected with CT26 cells	-	= expression in blood and LM	Therapeutic effect: ↓ tumor growth ↓ LM	[115]
Citrus(LMW pectin PYKTIN, Centrax Int.) 1–5% in drinking water	Nude mice engrafted with SW-480 cells	-	↓ expression	Therapeutic effect: ↓ tumor growth ↑ 5-FU effect ↑ apoptosis in tumor tissue	[116]
Citrus(modified pectin with alginate and *L. acidophilus*)	AOM-treated Balb/c mice	-	↓ expression in colonic crypts and blood vessels	Chemopreventive effect: ↓ precancerous lesions	[117]
Apple(modified pectin from red Fuji apples) 2.5–10% in pellet diet	DMH/DSS- treated ICS mice	-	↓ expression in serum ↓ galactose binding to SW-1116	Chemopreventive effect: ↓ colon inflammation ↓ tumorigenesis ↑ colonic EC apoptosis ↑ caspase-3 activation	[118]
*Smilax china L*. (pectin)	DSS-treated Balb/c mice	-	↓ Gal- 3/NLRP3 inflammasome interaction	Therapeutic/preventive effects: ↓ UC histo- pathological damage ↓ inflammatory mediators	[3]
Citrus(modified pectin, EcoNugenics) 0.1–25 mM	-	DLD-1	↓ extracellular expression	↓ cell migration	[81]
Citrus (LMW pectin PYKTIN, Centrax Int.) 0.625–10 mg/mL	-	SW-480	↓ expression	↓ cell proliferation ↑ cell cycle arrest ↓ EMT	[116]
Ginseng (HG-rich pectin) Ginseng(temperature-modified pectin)	-	HT-29 HT-29	ND	↓ cell proliferation ↑ cell cycle arrest ↓↓ cell proliferation ↑ apoptosis ↑ caspase-3 activation	[119]
			ND		
Ginseng (RG-I-4 pectic fragment)	-	HT-29	↓ rGal3- induced RBC agglutination ↓ binding to Jurkat cells	↓ cell adhesion ↓ ASF-induced cell aggregation	[120]
Apple(enzyme- modified, enriched in RG-I regions)	-	HCT 116 Caco-2 HT-29	↓ expression	↑ irinotecan effect ↓ cell viability ↑ apoptosis ↑ ROS ↓ LPS-induced inflammatory mediators	[121]
Sugar beet(enzyme- or alkali- modified pectin) 0.2–1 mg/mL	-	HT-29 DLD-1	Galactose/ arabinose- mediated	↓ cell proliferation ↑ apoptosis	[122]
Olive (heat- and acid-modified pectin) 1–10 mg/mL	-	Caco-2	↓ rGal3-induced RBC agglutination	↓ cell proliferation	[123]
Jaboticaba (pectin from fruit flour) 0.25–2 mg/mL	-	HCT116	↓ rGal3- induced RBC agglutination	↓ cell viability	[124]
Papaya (pectin from fruit pulp) 0.025–0.2%	-	HCT116 HT-29	↓ rGal3- induced RBC agglutination Gal-3 gene knockdown	↓ cell viability ↓ cell proliferation	[108,125]

Abbreviations: AOM, azoxymethane; ASF, asialofetuin; DMH, 1,2-dimethylhydrazine; DSS, dextran sodium sulfate; NDC, non-digestible carbohydrates; EC, epithelial cells; EMT, epithelial–mesenchymal transition; 5-FU, 5-fluorouracil; HG, homogalacturonan; LMW, low molecular weight; LPS, lipopolysaccharide; LM, liver metastases; NM, node metastases; RBC, red blood cells; rGal3, recombinant galectin 3; ROS, reactive oxygen species; UC, ulcerative colitis.

**Table 3 cancers-15-00052-t003:** Studies describing galectin-3-mediated effects of polyphenols in disease models.

Polyphenol	In Vivo Model	In Vitro Model	Galectin-3 Blocking	Main Results	Reference
Resveratrol 25–100 µM	-	Human ovarian cancer cell lines SKOV3 OVCAR-3	↓ expression (↑ mir- 424-3p/↓ NF- kB activation)	↓ cell proliferation ↑ apoptosis ↓ cell migration and invasion ↑ sensitivity to cisplatin	[140]
Quercetin oral intake 100 mg/kg	HFD fed ApoE−/− C57BL/6J mice	RAW264.7 macrophage cell line	↓ expression	↓ atherosclerotic lesions ↓ NLRP3 inflammasome activation	[141]
Berberine 5–25 µM	-	Ox-LDL- activated THP-1	↓ expression (↓ NF- kB/AMPK signaling)	↓ cell activation	[142]
Xanthohumol in drinking water 0.1%	HFD fed C57BL/6J mice	-	↓ expression in kidney and liver	↓ AGE in kidney ↓ 3-NT in liver	[143]
8- prenylnaringenin in drinking water 0.1%	HFD fed C57BL/6J mice	-	↓ expression in kidney and liver	↓ AGE ↓ 3-NT in liver and kidney	[143]

Abbreviations: AGE; advanced glycoxidation end products; EC, epithelial cells; HFD, high-fat diet; LM, liver metastases; MAP, modified apple polysaccharide; MCP, modified citrus polysaccharide; ND, node metastases; 3-NT, 3-nitrotyrosin; Ox-LDL, oxidized low-density lipoproteins.

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
