# Peer review of "Highlights on the Role of Galectin-3 in Colorectal Cancer and the Preventive/Therapeutic Potential of Food-Derived Inhibitors"

_cancers, 2022, doi:10.3390/cancers15010052_

Round 1
Reviewer 1 Report
Overall, this is a thorough and clear review. It is clear and well described, and the figures are informative. I have a few minor comments for the authors:
- some formatting errors, particular in the first paragraph.
- table 1 seems limited, is it necessary to have a table for this?
- the location of the protein isn't entirely clear. In the introduction, it is described as a glycan binding protein, but not specified that it can be ubiquitous. It is then discussed as extracellular or intracellular functions and only in the section of bowel is it described that it can have different subcellular locations. Maybe just make it clear in the introduction,
- look at formatting of the borders in table to ensure consistency.
Author Response
Overall, this is a thorough and clear review. It is clear and well described, and the figures are informative. I have a few minor comments for the authors:
- some formatting errors, particular in the first paragraph.
- table 1 seems limited, is it necessary to have a table for this?
- the location of the protein isn't entirely clear. In the introduction, it is described as a glycan binding protein, but not specified that it can be ubiquitous. It is then discussed as extracellular or intracellular functions and only in the section of bowel is it described that it can have different subcellular locations. Maybe just make it clear in the introduction.
- look at formatting of the borders in table to ensure consistency.
We thank the Reviewer for having appreciated our review and for the positive comments. According to her/his suggestions:
- the description of galectin-3 functions and localization has been clarified in the Introduction;
- Table 1 has been removed and its content has been included in the revised text (lines 220-229);
- the inaccuracies have been fixed.
Reviewer 2 Report
Major concerns:
1. In general, the readability of this review is quite low. Examples for the first few pages are detailed below. The authors are encouraged to get editing help from someone with full professional proficiency in English. Some paragraphs should be combined for example, line 67-87. Those paragraphs are broken up too much. “Furthermore” appears seven times in this article. The usage of “nevertheless” or “nonetheless” should be removed as those adverbs do not add any important meaning to the sentence.
2. Overall, the structure of this manuscript needs to be revised thoroughly as there are many reviews on Gal-3 in cancers previously published. As it reads, there are 11 pages of Gal-3 in cancers and 6-7 pages concerning the effect of “food-derived inhibitors” as natural Gal-3 inhibitors. The latter seems to be unique to this review and the authors should edit the first part to be more concise (reduce page numbers) and focus primarily on the latter part.
3.All figure legends need to be revised thoroughly to include more details for making them self-explanatory.
For example,
1. Figure 1. A) It needs to be stated that Gal-3 is the only chimeric type. What is the tandem repeat type? What is the color difference between the same shaped domains in green and in blue and what does this color scheme mean? B) Panels a & b: there is no description for C-terminal. What is the double headed curved arrow connecting the two panels? Why is it placed on top of the two panels but not in between?
2. Figure 2. What is crypt? What is gland? Base towards the surface à basal to apical
3. Figure 3. What is PRR? What are the two Y-shaped things? What is the curved line connecting them? What is the difference between the solid line arrow from fibers versus the broken arrow from the polyphenol?
4. In Table 2 title, replace IBD with UC as no other IBD is listed.
5. In Table 3 the label of the far-left column should be pectin source and modification.
6. Because non-digestible is abbreviated as ND, the not determined should be replaced with – to avoid any confusion.
Examples of readability problems
Under Simple summary
1. Line 13 sentence reads awkward.
2. Line 14 are a major concern à are major concerns.
3. Line 15 Animportant à an important
4. Line 16 “Chinese box” or a “Matrisoska doll” à Inappropriate descriptions that are not suited to describe a protein function.
5. Line 17 Clearcut relationship à awkward expression. It should be “clear cut” or “clear-cut” but this is a subjective term and not scientific.
6. Line 22 tumor spread à spread of cancer cells
Under Abstract
1. Line 26 therapy resistance à resistance in therapy
2. Line 120 Some plant-origin food derived molecules à some molecules such as XXX and YYY in plant-based foods. Provide examples.
3. Line 154 -157 What is disease activity? Starting the sentence with nevertheless and having regardless in the middle of the sentence does not flow well.
4. Line 169 levels were associated with the disease and a more malignant biological behavior.
“biological behavior” sounds odd. This is about disease.
5. Line 199 “combined model” What does model mean?
6. Line 220 extracellular-membranous à sounds awkward
7. Line 225 Provide a couple of examples for cell surface signaling receptors.
8. Line 237 Therefore, a Gal-3 targeting may have a combined effect on EGFR-targeted therapy [82]. à What is “a” Gal-3 targeting?
9. Line 245 By directly linking MUC1 à linking MUC1 to what?
10. Line 282 “bad molecule” is just not scientific.
There are more.
Author Response
- In general, the readability of this review is quite low. Examples for the first few pages are detailed below. The authors are encouraged to get editing help from someone with full professional proficiency in English.
Some paragraphs should be combined for example, line 67-87. Those paragraphs are broken up too much. “Furthermore” appears seven times in this article. The usage of “nevertheless” or “nonetheless” should be removed as those adverbs do not add any important meaning to the sentence.
According to the Reviewer’s requests, the English language has been extensively edited and the style of the manuscript has been improved.
- Overall, the structure of this manuscript needs to be revised thoroughly as there are many reviews on Gal-3 in cancers previously published. As it reads, there are 11 pages of Gal-3 in cancers and 6-7 pages concerning the effect of “food-derived inhibitors” as natural Gal-3 inhibitors. The latter seems to be unique to this review and the authors should edit the first part to be more concise (reduce page numbers) and focus primarily on the latter part.
We agree with the Reviewer that there are a number of reviews on galectin-3 and cancer. Our review, however, specifically focuses on the role of galectin-3 in colorectal cancer (CRC) and in intestinal inflammatory diseases that represent a risk factor for CRC development. In addition to describing the direct effects of galectin-3 on colon cancer cell grow and spread, the manuscript also highlights the important contribution of this protein to the generation/sustainment of the immunosuppressive tumour microenvironment through its indirect effects on immune cells. Finally, we addressed the still poorly investigated issue of food-derived galectin-3 inhibitors as potential/novel strategy for CRC prevention and as adjuvant to therapy. In our opinion, all these aspects make the review novel and unique. For this reason, we prefer to maintain the original structure.
- All figure legends need to be revised thoroughly to include more details for making them self-explanatory.
For example,
- Figure 1. A) It needs to be stated that Gal-3 is the only chimeric type. What is the tandem repeat type? What is the color difference between the same shaped domains in green and in blue and what does this color scheme mean? B) Panels a & b: there is no description for C-terminal. What is the double headed curved arrow connecting the two panels? Why is it placed on top of the two panels but not in between?
- Figure 2. What is crypt? What is gland? Base towards the surface à basal to apical
- Figure 3. What is PRR? What are the two Y-shaped things? What is the curved line connecting them? What is the difference between the solid line arrow from fibers versus the broken arrow from the polyphenol?
- In Table 2 title, replace IBD with UC as no other IBD is listed.
- In Table 3 the label of the far-left column should be pectin source and modification.
- Because non-digestible is abbreviated as ND, the not determined should be replaced with – to avoid any confusion.
As a follow up to the Reviewer’s comments, all Figure legends and Tables have been revised. The changes are highlighted in yellow.
Examples of readability problems
Under Simple summary
- Line 13 sentence reads awkward.
- Line 14 are a major concern à are major concerns.
- Line 15 Animportant à an important
- Line 16 “Chinese box” or a “Matrisoska doll” à Inappropriate descriptions that are not suited to describe a protein function.
- Line 17 Clearcut relationship à awkward expression. It should be “clear cut” or “clear-cut” but this is a subjective term and not scientific.
- Line 22 tumor spread à spread of cancer cells
Under Abstract
- Line 26 therapy resistance à resistance in therapy
- Line 120 Some plant-origin food derived molecules à some molecules such as XXX and YYY in plant-based foods. Provide examples.
- Line 154 -157 What is disease activity? Starting the sentence with nevertheless and having regardless in the middle of the sentence does not flow well.
- Line 169 levels were associated with the disease and a more malignant biological behavior.
“biological behavior” sounds odd. This is about disease.
- Line 199 “combined model” What does model mean?
- Line 220 extracellular-membranous à sounds awkward
- Line 225 Provide a couple of examples for cell surface signaling receptors.
- Line 237 Therefore, a Gal-3 targeting may have a combined effect on EGFR-targeted therapy [82]. à What is “a” Gal-3 targeting?
- Line 245 By directly linking MUC1 à linking MUC1 to what?
- Line 282 “bad molecule” is just not scientific.
There are more.
According to the Reviewer's comments, we have removed the inappropriate expressions and fixed the inaccuracies. The style has been checked and improved throughout the manuscript, as also stated in the answer to comment 1. Information about signalling receptors has been included in the revised text (lines 233-238).
Reviewer 3 Report
The present review manuscript titled “Highlights on the Role of Galectin-3 in Colorectal Cancer and the Preventive/Therapeutic Potential of Food-Derived Inhibitors” by Aureli et al. is novel and very well written. The present manuscript highlights the role of Galectin-3 in the development of colorectal cancer and its prevention/therapy with food-based Galectin-3 inhibitors. Overall, the quality of the manuscript is excellent. However, some corrections are needed to improve the quality of the manuscript. My comments are as follows.
Comment 1. First of all, the role of Galectin-3 in the progression of colorectal cancer should be discussed in a separate section in detail. Under this section, sections 2 and 3 should be discussed.
Comment 2. Patents on this evidence should also be discussed in the manuscript.
Comment 3. The future direction and conclusion sections should be separately discussed. However, the conclusion is poor. The authors should revise this section.
Comment 4. I agree with the authors that the food-derived inhibitors played a significant inhibitory effect on colorectal cancer. Therefore, the authors should discuss some clinically prescribed food-derived inhibitors in the manuscript.
Comment 5. Kindly revise the references as per the journal requirement.
Author Response
The present review manuscript titled “Highlights on the Role of Galectin-3 in Colorectal Cancer and the Preventive/Therapeutic Potential of Food-Derived Inhibitors” by Aureli et al. is novel and very well written. The present manuscript highlights the role of Galectin-3 in the development of colorectal cancer and its prevention/therapy with food-based Galectin-3 inhibitors. Overall, the quality of the manuscript is excellent. However, some corrections are needed to improve the quality of the manuscript. My comments are as follows.
Comment 1. First of all, the role of Galectin-3 in the progression of colorectal cancer should be discussed in a separate section in detail. Under this section, sections 2 and 3 should be discussed.
We thank the Reviewer for the positive comments. According to her/his suggestion, we have reorganized the structure of the manuscript. Specifically, previous sections 2 and 3 (now sub-sections 2.1, 2.2 and 2.3) have been combined in a more general section (section 2: “Galectin-3 in colorectal cancer development and progression”) in which we described i) the role of galectin-3 in intestinal inflammation and colorectal cancer (human studies), ii) the direct effects of this protein on cancer cells and iii) the indirect immune-mediated effects. The changes are highlighted in yellow in the revised version.
Comment 2. Patents on this evidence should also be discussed in the manuscript.
As a follow-up to the Reviewer’s comment, we have discussed the current clinical studies which aim to validate galectin-3 as a biomarker for colorectal cancer and galectin-3 antagonists as anticancer molecules. This information has been included in the revised manuscript (lines 211-217).
Comment 3. The future direction and conclusion sections should be separately discussed. However, the conclusion is poor. The authors should revise this section.
According to the Reviewer’s suggestion, the “Conclusion” section has been expanded and discussed separately from the “Future directions” section.
Comment 4. I agree with the authors that the food-derived inhibitors played a significant inhibitory effect on colorectal cancer. Therefore, the authors should discuss some clinically prescribed food-derived inhibitors in the manuscript.
Only a few human trials with pectin-based molecules have been developed to date, despite the large number of epidemiologic and preclinical studies arguing for a protective role of dietary fibers in cancer prevention and therapy. The modified citrus pectin (MCP) has been approved as a dietary supplement with antioxidant and immunomodulatory properties (PectaSol-C, NCI Code C103178). Currently it is under clinical study for a small number of cancer types such as prostate cancer and melanoma. Concerning the polyphenols, a wide range of clinical trials have been registered in which polyphenolic compounds (e.g. curcumin, genistein, fisetin, silybin, epigallocatechin gallate) are being tested as preventive and anticancer medicines, mostly in combination with other drugs, in different cancer types, including CRC and precancerous colon adenomatous polyposis. Dietary supplements containing these compounds are indicated as anti-inflammatory and anti-oxidant substances.
As suggested by the Reviewer, this information has been included in the revised text (lines 498-508 and lines 517-521).
Comment 5. Kindly revise the references as per the journal requirement.
According to the Reviewer’ suggestion, we revised the reference format.
Round 2
Reviewer 2 Report
It seems that authors have answered most of my points.
Reviewer 3 Report
The authors addressed all the comments very systematically. I am satisfied with their correction. I don't have further comments.